# Dietary Assessment Tools and Metabolic Syndrome: Is It Time to Change the Focus?

**DOI:** 10.3390/nu14081557

**Published:** 2022-04-08

**Authors:** Helen Chauhan, Regina Belski, Eleanor Bryant, Matthew Cooke

**Affiliations:** 1School of Health Sciences, Swinburne University of Technology, Hawthorn, VIC 3122, Australia; hchauhan@swin.edu.au (H.C.); rbelski@swin.edu.au (R.B.); 2School of Allied Health, Human Services and Sport, La Trobe University, Melbourne, VIC 3086, Australia; 3Division of Psychology, University of Bradford, Bradford BD7 1DP, UK; e.j.bryant@bradford.ac.uk

**Keywords:** metabolic syndrome, carbohydrate, processed, meal timing, diet, assessment tool

## Abstract

Metabolic syndrome (MS) is associated with a range of chronic diseases, for which lifestyle interventions are considered the cornerstone of treatment. Dietary interventions have primarily focused on weight reduction, usually via energy restricted diets. While this strategy can improve insulin sensitivity and other health markers, weight loss alone is not always effective in addressing all risk factors associated with MS. Previous studies have identified diet quality as a key factor in reducing the risk of MS independent of weight loss. Additionally, supporting evidence for the use of novel strategies such as carbohydrate restriction and modifying the frequency and timing of meals is growing. It is well established that dietary assessment tools capable of identifying dietary patterns known to increase the risk of MS are essential for the development of personalised, targeted diet and lifestyle advice. The American Heart Association (AHA) recently evaluated the latest in a variety of assessment tools, recommending three that demonstrate the highest evidence-based and clinical relevance. However, such tools may not assess and thus identify all dietary and eating patterns associated with MS development and treatment, especially those which are new and emerging. This paper offers a review of current dietary assessment tools recommended for use by the AHA to assess dietary and eating patterns associated with MS development. We discuss how these recommendations align with recent and novel evidence on the benefits of restricting ultra-processed food and refined carbohydrates and modifying timing and frequency of meals. Finally, we provide recommendations for future redevelopment of these tools to be deployed in health care settings.

## 1. Introduction

The clustering of metabolic disturbances linked to cardiovascular disease was first described by Swedish physician Eskil Kylin in 1923 [1]. In 1947, Jean Vague noted upper-body obesity as the phenotype most commonly associated with the metabolic abnormalities linked to type 2 diabetes and cardiovascular disease [2]. However, it was not until 1975 that the term ‘metabolic syndrome’ (MS) was first used in the medical literature by Hermann Haller [3], with Gerald Reaven later proposing in 1988 that insulin resistance was the common feature of this syndrome [4]. Whilst there are various definitions of MS, the most favoured was developed by the US National Cholesterol Education Program (NCEP) Adult Treatment Panel III (ATP III) in 2001. This definition incorporates the key diagnostic criteria of hyperglycaemia/insulin resistance, visceral obesity, atherogenic dyslipidaemia and hypertension, and allows the use of readily available anthropomorphic, hemodynamic and blood assessments to diagnose MS based on the presence of three out of the five aforementioned criteria [5].

The US National Health and Nutrition Examination Survey (NHANES) (1999–2002) estimated the age-adjusted prevalence of MS in United States adults aged 20 years or greater to be between 34.6% and 39.1% [6]. Unpublished data from the AusDiab study (1999–2000) suggested that prevalence of MS in Australian adults was between 23.9% and 26% [7]. More recent NHANES data suggest that the overall prevalence of MS did not significantly increase in the period 2011–2016; however, statistically significant increases in prevalence amongst younger adults (aged 20–39), women and Asian and Hispanic individuals were still noted [8]. Globally, MS is estimated to be 3-fold more common than type 2 diabetes, with prevalence estimated at one-quarter of the world’s population [9]. Individuals with MS have a 5-fold higher risk of developing type 2 diabetes and a 3-fold higher risk of developing cardiovascular disease [10]. In the period 2015–2016, an estimated 8.9% ($10.4 billion) of total disease expenditure in the Australian health system was attributed to cardiovascular diseases and an estimated 2.3% ($2.7 billion) to diabetes [11]. To translate this to the health care costs of an individual, a 2009 report suggested that for each risk factor a person develops (i.e., obesity, high blood pressure, etc.), health care costs can increase nearly 1.6-fold (approx. $2,000 USD). For each additional risk factor, those costs can rise by an average of 24% [12].

## 2. The Aetiology of Metabolic Syndrome

The aetiology of MS is beyond the scope of this review and a more detailed discussion can be found elsewhere [13]. Briefly, insulin resistance and the underlying pathophysiological mechanisms that contribute to its development have been identified as major contributors to the development of MS [14,15]. Obesity has historically been regarded as one of the leading causes of MS, however, whether weight gain precedes hyperinsulinemia and insulin resistance or whether hyperinsulinemia drives weight gain and metabolic disease is often debated [16]. Evidence from studies investigating the use of exogenous insulin to manage both type 1 and type 2 diabetes has demonstrated that intensive control of blood glucose through increased insulin use resulted in weight gain [17] even when caloric intake was reduced [18]. Moreover, a recent meta-analysis of 60 longitudinal studies and randomised clinical trials revealed temporal sequencing between fasting insulin, body mass index and systemic inflammation [19]. Their findings showed that changes in fasting insulin preceded changes in weight gain and did not support the assertion that obesity comes first before elevated fasting insulin levels and disease development. Consequently, the authors suggested that hyperinsulinemia may be the driver of adverse health consequences, rather than weight (obesity) itself [19]. While it is well acknowledged that obesity and elevated body fat levels alone can negatively impact the body’s physiology and internal homeostasis, this area of ongoing research does raise questions regarding the most effective lifestyle and dietary approaches for preventing and reversing MS, with weight loss potentially becoming not the only target. Despite this, current guidelines and interventions still primarily focus on weight loss, meaning that moving beyond this would be timely.

## 3. Diet and Lifestyle Interventions—Moving beyond Weight Loss

The traditional diet and lifestyle interventions used to treat MS involve eating less and moving more. This approach aligns with the current guidelines for the management of overweight and obesity by the Australian National Health and Medical Research Council, which recommends caloric restriction, increased physical activity and behaviour modification to create a 2500 kilojoule energy deficit and achieve a 5–10% weight reduction [20]. While these guidelines can be successful, especially in the short term, prioritising weight loss as the primary goal for treating MS can be limiting. Firstly, these dietary and lifestyle approaches do not take into account the concept that energy input and expenditure are interdependent and complex, rather than ‘one simply influences the other’. It is evident that energy balance is controlled by multiple feedback mechanisms that help maintain body weight within a narrow range [21]. Subsequently, many individuals that solely focus on an energy deficit model to lose weight often fail and tend to gain the same (and more) weight back [22].

In addition, the focus on weight loss inherently means targeting those with a higher BMI. This focus could increase the likelihood of overlooking at-risk populations with MS that fall within the normal-weighted BMI category. Identified as metabolically obese but normal weight (MONW), this subgroup of normal-weight individuals displaying obesity related phenotypic characteristics are at higher risk for type 2 diabetes and cardiovascular diseases [23]. A recent analysis of mortality rates in different weight categories amongst 12,047 US adults with and without MS found that the prevalence of MS in normal-weight individuals was 8.6% [24]. Interestingly, multivariate risk analysis demonstrated that these individuals displayed the highest mortality rate [24]. Furthermore, recent data from the UK and US suggest the application of different BMI cut-off points for certain ethnic groups such as South Asian, Arab, Chinese and Black populations that are at higher health risk at lower BMI thresholds [23,25]. Collectively, these observations indicate that MONW individuals should also be targeted for lifestyle intervention.

With new and emerging research in this area, there is growing support for moving the focus away from weight loss, typically by caloric restriction, and directing our attention toward the diet or eating pattern that is most appropriate for treating MS. While this topic is also highly debated, the current standard dietary recommendations for treating MS include promoting low intakes of saturated and trans fats, reducing consumption of simple sugars and increasing intakes of fruit, vegetables and whole grains [26]. Although these eating patterns are typically captured within the current dietary assessment tools, alternative and emerging dietary patterns/strategies that have demonstrated effectiveness for treatment and management of MS such as reducing overall carbohydrate intake, ultra-processed foods and timing of food intake are not [27,28,29].

### 3.1. Carbohydrate Restriction

The current Dietary Guidelines for Americans [30] recommend that half of calories consumed be derived from carbohydrates in an effort to limit the intake of dietary fat. US government data suggest that, in terms of percentage of total energy intake, consumption of carbohydrates has increased by 30% whilst consumption of fat has decreased by 25% since 1965 [31]. Given the association between MS and insulin resistance, overconsumption of carbohydrates, especially those known to induce large spikes in insulin levels, could be linked to MS development [31].

Contemporary evidence suggests that restricting dietary carbohydrates is an effective means of targeting a range of risk factors associated with MS with a single modification to dietary intake [32]. Clinical trials have shown that diets low in sugar and refined carbohydrates, while high in whole foods and healthy fats, can reduce atherosclerotic cardiovascular disease risk in overweight and obese adults [33]. This approach has demonstrated effectiveness when uniformly restricting carbohydrate intake in short-term studies, e.g., to 12% of total calories [32] and in long-term programs that adapted dietary intake of carbohydrates to suit the personal circumstances of participants [34]. Improvements in clinical features of MS such as elevated blood pressure, hyperglycaemia, weight and lipid profiles have also been shown to be both achievable and sustainable in the long term through moderate carbohydrate restriction (<120 g per day) in community settings through the avoidance of sugary and starchy foods such as breakfast cereals, bread, pasta and rice [35]. A dietary assessment tool that identifies high-starch foods, which might feature in every meal, and high-sugar foods provides the opportunity to discuss options for substituting with lower-carbohydrate alternatives.

### 3.2. Minimising Ultra-Processed Food

The term “ultra-processed food” (UPF) was coined in epidemiological studies that found an association between UPF consumption and a range of chronic diseases including MS, irritable bowel disease and cancer [36]. The hedonistic qualities of UPF are thought to encourage reward-driven eating and play a role in overriding biological controls of appetite and satiety [37], which impacts weight and health [38]. The extent to which food has been processed plays a part in its nutritional quality, so categorising food by levels of processing provides valuable information beyond defining foods simply by food group [39]. The NOVA food classification system developed at the University of Sao Paulo, Brazil, classifies foods into four categories: unprocessed or minimally processed (e.g., fresh meat and vegetables or pasteurised milk), processed culinary ingredients (e.g., sugar or oil), processed food (e.g., canned fish and fruit in syrup), and ultra-processed food (e.g., sweet or savoury packaged snacks and pre-prepared frozen meals). Critics of the NOVA classification system argue that it simply identifies food that are likely high in sugar, fat and salt [40] and therefore adds nothing new. While this is in part true, identifying and classifying these types of foods are helpful for a number of reasons. Firstly, evidence shows that the highly addictive potential of ultra-processed foods is related to their added sugar content [41]. Secondly, the rewarding properties of high-fat foods and their links to overconsumption appear to occur when fats and carbohydrates are consumed together [42]. Thus, while traditional dietary advice and definitions of a healthy diet have focused on specific nutrients in food, the NOVA classification system focuses on the extent and nature of food processing and identifying such foods or dietary patterns (via dietary assessment tools) could help further mitigate known factors that contribute to the development of MS.

### 3.3. Meal Timing and Frequency

Dietary patterns are considered central to combating metabolic diseases. However, there is an emerging acceptance that meal timing may be as or more important than the amount or type of food consumed [43], and combining these approaches may elicit greater benefits [44]. Surveys conducted between 1977 and 2006 demonstrated that daily “eating occasions” increased in adults and children, with energy intake, particularly from snacking, increasing and the time between “eating opportunities” decreaseing from 3.5 to 3 h [45]. Circadian rhythms, the 24 h cycles that are part of the body’s internal clock, run in the background to carry out essential functions and processes, including weight regulation. The central clock of the suprachiasmatic nucleus of the hypothalamus controls many circadian rhythms, as well as clocks located in other brain regions and most peripheral tissues [46]. Chronic circadian rhythm disruption, such as shift work or repetitive late night snacking, is a risk factor for metabolic diseases and both human and animal studies have demonstrated that time-restricted feeding can provide protection from circadian rhythm-induced metabolic disturbances [44]. Food is a non-photic stimulus that can reset the circadian rhythm by predominantly influencing the peripheral clocks and the timing of ‘when’ the majority of calories are eaten is an important factor [47]. For example, consuming the majority of calories in the evening or at night is associated with a higher risk of non-alcoholic fatty liver disease, whilst eating main meals earlier in the day is associated with decreased risk for hepatic steatosis [48]. Alternatively, it may involve restricting your window of eating during the day. Indeed, both early time-restricted eating (from 8:00 a.m. to 5:00 p.m.) and delayed time-restricted eating (from 12:00 p.m. to 9:00 p.m.) improved glycaemic response to a test meal, with the early pattern also leading to a decrease in fasting glucose [49]. Given the profound benefits on strategic meal timing/frequency on chronic disease risk factors such as MS, it is perhaps time to consider including such practices within dietary assessment tools, and, in doing so, help facilitate discussion with an individual about which timing strategy may best suit them.

## 4. Existing Tools for Assessing Dietary Risk for Metabolic Syndrome

The diagnosis of MS offers an opportunity to address several risk factors within an individual, especially those related to the development of cardiovascular disease, type 2 diabetes, and chronic kidney disease [14]. A wide variety of screening tools are available to assess dietary intake, with the most appropriate being selected based on the purpose for which dietary information is being collected. For clinicians and other members of a health care team who focus on preventing and treating chronic diseases in diverse clinical settings, these tools must be able to be administered quickly, be valid, and reflect ‘up-to-date’ dietary guidelines. In 2020, the American Heart Association published a scientific statement detailing the importance of point-of-care dietary assessment and incorporating screening or assessment tools to support discussions between clinicians and patients about diet quality to improve the management of diet-related chronic disease [50]. The authors reviewed 15 tools developed in the past 10 years against a set of theory and practice-based validity criteria relevant to routine clinical practice and identified 3 that met the greatest number of criteria. These were the Mediterranean Diet Adherence Screener (MEDAS) and its variations, the modified Rapid Eating Assessment for Participants (REAP) and the modified version of the previously validated Starting the Conversation tool. These tools were highlighted by the authors as potentially helpful to enable clinicians to identify key dietary risk factors of MS and to facilitate discussions about dietary change [50].

### 4.1. MEDAS

The Mediterranean diet is an eating approach associated with the increased consumption of fresh, whole foods (olive oil, nuts, seeds, vegetables, fruit, and fish) and a decrease in red meat and processed foods. This way of eating carries particular favour and has been shown to be beneficial when combined with general lifestyle modification [51]. Moreover, this dietary approach has long been associated with low cardiovascular disease risk in adult populations, and prospective studies, including clinical trials, have demonstrated that adherence to the Mediterranean diet is associated with reduced risk of MS [52]. A meta-analysis of nine controlled studies involving 1178 patients with type 2 diabetes showed that a Mediterranean-style diet led to greater reductions in haemoglobin A1c, fasting blood glucose, fasting insulin, BMI and body weight compared to other approaches such as low-fat diets [53].

The original 14-item MEDAS tool was developed and validated in the Spanish Prevención co Dieta Mediterránea (PREDIMED) study, where higher MEDAS scores were associated with healthier lipid profiles, lower weight and waist circumference, lower fasting glucose and reduced 10 year coronary artery disease risk [54]. The tool has been validated using smaller cohorts in Germany and the UK, although a UK study of 96 adults with high cardiovascular risk did not observe an association between MEDAS score and cardiovascular risk [55]. The tool has been adapted slightly to create the Mediterranean Eating Pattern for Americans, where a higher score was shown to be positively associated with “higher diet quality” without being significantly associated with lower BMI [56].

The MEDAS tool centres key assumptions regarding the components of a Mediterranean diet that are considered beneficial and was designed to assess compliance with this dietary approach. However, when used in a broader context to evaluate diet, it has a range of limitations. For example, whilst the use of olive oil in cooking may be considered preferable to the use of processed vegetable oils, the emphasis on high levels of olive oil consumption (≥4 tablespoons per day) may not be compatible with many dietary patterns. Similarly, the tool endorses a higher consumption of fruit (3 or more servings) than vegetables (2 or more servings). Whilst the consumption of vegetables and fruit has been inversely associated with incidence of MS [57], a recent randomised control study yet to undergo peer review has shown that a 6 month-long, high-fruit diet (4 or more servings a day) had a statistically significant detrimental effect on BMI, grade of steatosis, liver enzymes, dyslipidaemia, fasting blood glucose and insulin resistance in participants with existing non-alcoholic fatty liver disease. In contrast, participants consuming fewer than 2 portions of fruit a day demonstrated significant improvements in BMI, steatosis, lipid profile and insulin resistance [57]. The tool does not differentiate between butter, a minimally processed food, and margarine, which is ultra-processed, and endorses consuming 1–2 glasses of wine per day 7 days a week instead of other alcoholic beverages. Whilst there is evidence that moderate alcohol consumption is associated with reduced incidence of MS, it is not clear that wine is more beneficial than other forms of alcohol [58], and a more recent review suggests that the overall harm of alcohol consumption far outweighs any possible benefit from very limited consumption [59]. The strength of the tool is its overall emphasis on consuming fresh, unprocessed, or minimally processed food and not consuming sugar-sweetened beverages. Nonetheless, this is outweighed by the semantics of the definition of the Mediterranean diet. The tool’s emphasis on minimising saturated fat intake does not reflect the most recent evidence [60].

### 4.2. REAP-S

Dietary Guidelines for Americans were first published in 1980 and have been updated periodically since. The most recent edition [30] sets out a range of acceptable macronutrient ranges, lower in fat and higher in carbohydrates relative to the average American diet prior to the implementation of the original 1980 guidelines. The original 27 to 31-item REAP tool was designed to rapidly assess diet quality, particularly in lower-income populations. A shortened 16 item version (REAP-S) which has less emphasis on dietary fat has been shown to be moderately correlated with the Healthy Eating Index 2010 [61]. It is regarded as a useful tool for assessing dietary quality and eating habits in relation to the US dietary guidelines and is used to inform dietary counselling to patients [62].

The REAP-S tool goes some way to highlight high-sugar foods and drinks, but the thresholds are high (e.g., ≥16 oz of sugar-sweetened beverages a day and eating sweets or cakes ≥ twice a day). The baseline threshold for both vegetable and fruit consumption is 2 servings per day for each and includes fruit juices, a source of dietary sugar associated with fatty liver disease [63]. The tool does not differentiate between minimally and highly processed foods, for example grouping brown rice with whole grain breakfast cereals and crackers and rating low-fat chips, crackers and processed meats more highly than regular chips, crackers and processed meats.

### 4.3. Starting the Conversation Tool

The Starting the Conversation tool was designed specifically for use by workers without specialist training in busy clinical settings to assist in identifying atherogenic dietary patterns. The tool has been found to be robust and valid across diverse participants [64]. This tool also prioritises a high intake of vegetables and fruit, allocating the highest score to consuming five or more servings per day of each. The tool’s scoring system implies that low-fat chips, crackers, desserts and sweet foods are acceptable, even though they are highly processed and high in refined carbohydrates. The strength of the tool is that its threshold for consuming fast food is less than once a week and sugar-sweetened beverages less than once a day.

Despite the strengths of the aforementioned tools, there is a hesitancy amongst health care practitioners to utilise such tools and provide interventions such as basic dietary counselling. This hesitancy has been attributed to barriers experienced by practitioners such as a lack of nutrition knowledge and training, and the time it takes to undergo dietary assessment [50]. Subsequently, there is a tendency for clinicians to focus on individual components of MS, such as blood pressure or triglycerides, that are amenable to drug therapies and easier to administer, rather than a lifestyle change program [65].

### 4.4. Evaluation of MEDAS, REAP-S and Starting the Conversation Tool

The three rapid screening tools have been evaluated by the authors to consider the extent to which they reflect more recent evidence and understanding of factors contributing to the development of MS: the principles of lowering carbohydrate intake, minimising ultra-processed foods and managing meal timing and frequency. A summary of these findings is presented in Table 1. All tools demonstrate limited capacity to clearly differentiate between high- and low-carbohydrate foods. In most instances, this is likely a reflection of the broad nature of the questionnaire items and lack of detail to identify specific high- and low-carbohydrate foods. For example, the REAP-S tool includes “eating more than 4 meals per week from sit down or take-out restaurants”, which does not provide any information about the type of food or macronutrient consumed. While all three tools include questions relating to fruit and vegetable consumption, they again fail to differentiate between higher-carbohydrate items such as potatoes, corn and bananas versus lower-carbohydrate items such as leafy greens and berries. A strength of the three tools is the overall focus on fresh food consumption, which may correspond to reduced consumption of UPFs. However, without clear differentiation regarding types of fresh food, the tools could miss an opportunity to identify unhealthy habits. For example, the emphasis on certain dietary patterns such as lower dietary fat and meat intake and reduced saturated and animal fats potentially fails to capture the alternatives such a higher carbohydrate intake and greater consumption of ultra-processed vegetable oils and margarines. In addition, there is no differentiation between foods that are prepared at home or in a restaurant from raw ingredients, versus pre-prepared meals. The greatest area of weakness of all three tools lies in their failure to identify meal timing or frequency dietary habits, though the REAP-S tool does highlight skipping breakfast as a negative attribute. Moreover, the tools provide no opportunity to identify relevant risk factors such as frequent snacking and late-night eating. As highlighted in Table 1, there are a lot of instances where more details are required. This predominantly centres around lack of discrimination between high- and low-carbohydrate foods and the focus on some ‘healthy’ dietary habits, which could potentially miss other ‘unhealthy’ habits that are known risk factors of MS development. Addressing these limitations could in turn change the degree of relevance of each tool to each of the three categories and help ensure that each tool better reflects more recent evidence regarding dietary patterns contributing to the development of MS.

## 5. Conclusions

It is well established that MS represents a highly significant global health challenge. The continued focus on weight loss as the driver for reversing MS is unhelpful, as it is often accompanied by dietary and lifestyle advice that is not based on up-to-date evidence and does not address the increased mortality risk of normal-weight individuals with MS. At present, dietary assessment tools recommended by the AHA for screening diets of those at risk of and with MS are somewhat outdated, and all lack the ability to screen for and identify more emerging dietary risk behaviours such as excessive refined carbohydrate consumption, the consumption of UPF and meal timing. Addressing these limitations in the AHA-recommended tools would enable a more thorough assessment of dietary behaviours that influence the development/worsening of MS that is better aligned with current evidence. If a short, simple, validated tool that incorporates up-to-date evidence were made available, this would hopefully alleviate hesitancy in using the tool by better capacitating clinicians to quickly and accurately identify dietary risk factors and support the subsequent implementation of dietary and lifestyle interventions to better prevent and manage MS. Further research is required to develop and pilot a tool that can be used in health care settings to guide dietary assessment and recommendations, implement lifestyle change goals and measure the impact of such changes on markers of MS.

## Figures and Tables

**Table 1 nutrients-14-01557-t001:** Analysis of the relevance of screening tools to more recently proposed dietary approaches.

Tool	Questionnaire Items	Relevance to Alternative Approaches		
		Low Carbohydrate	Minimising UPFs	Meal timing and Frequency
Starting the Conversation	Frequency of intakes over the previous few months:			
	Fast food meals or snacks per month?	More detail required	✓	✕
	Servings of fruit per day?	More detail required	✓	✕
	Servings of vegetables per day?	More detail required	✓	✕
	Regular sodas, juices or other sugary beverages per day?	✓	✓	✕
	Servings of beans, nuts, chicken or fish per week?	Include all protein sources	More detail required	✕
	Regular snack chips or crackers per week?	✓	✓	✕
	Desserts and other sweets per week?	✓	More detail required	✕
	Use of butter or meat fat?	✕	✕	✕
REAP-S	In an average week, how often do you:			
	Skip breakfast?	✕	✕	More detail required
	Eat ≥ 4 meals from sit-down or take-out restaurants?	More detail required	More detail required	✕
	Eat <2 servings of fruit a day?	More detail required	More detail required	✕
	Eat <2 servings of vegetables a day?	More detail required	More detail required	✕
	Eat >8 oz meat, chicken, turkey or fish per day?	Include all protein sources	Include all protein sources	✕
	Eat regular processed meats instead of low-fat processed meats?	✕	✕	✕
	Eat fried foods such as chicken, fish, French fries, plantains, tostones, yukka?	✕	✕	✕
	Eat regular potato chips, nacho chips, corn chips, crackers or regular popcorn instead of unsalted nuts or air popped popcorn?	Limited relevance	✓	✕
	Eat sweets such as cake, cookies, donuts, muffins, chocolate and candies ≥2 times per day?	✕	More detail required	✕
	Drink ≥16 oz of non-diet soda, fruit drink/punch, or Kool-Aid a day?	Requires lower threshold	Requires lower threshold	✕
	Usually shop and cook rather than eating sit-down or take-out restaurant food?	✕	✕	✕
	Usually feel well enough to shop or cook	✕	✕	✕
	How willing are you to make changes in your eating habits to be healthier	✕	✕	✕
MEDAS	Do you use olive oil as the principal source of fat for cooking?	✓	✓	✕
	How much olive oil do you consume per day?	✕	✕	✕
	How many servings of vegetables do you consume per day?	More detail required	✓	✕
	How many servings of red meat, hamburger or meat products do you consume per day?	✓	✓	✕
	How many servings of butter, margarine or cream so you consume per day?	✕	✕	✕
	How many sugar-sweetened beverages do you drink per day?	✓	More detail required	✕
	How much wine do you drink per week?	✕	✕	✕
	How many servings of pulses do you consume per week?	✓	✓	✕
	How many servings of fish/shellfish do you consume per week?	✓	✓	✕
	How many times do you consume commercial sweets or pastries (not homemade) such as cakes, cookies, biscuits or custard?	✓	✓	✕
	Do you prefer to eat chicken, turkey or rabbit instead of beef, pork, hamburgers or sausages?	✕	✕	✕
	How many times per week do you consume cooked vegetables, pasta, rice or other dishes prepared with a sauce of tomato, garlic, onions or leeks sauteed in olive oil?	✕	✕	✕

UPFs: Ultra-Processed Foods; ✓ Questionnaire item is relevant to alternative dietary approach; ✕ Questionnaire item is not relevant to alternative dietary approach; REAP-S: Rapid Eating Assessment for Participants—shorterned; MEDAS: Mediterranean Diet Adherence Screener.

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
