# Peer review of "Dietary Assessment Tools and Metabolic Syndrome: Is It Time to Change the Focus?"

_nutrients, 2022, doi:10.3390/nu14081557_

Round 1

Reviewer 1 Report

Overall: This paper reviewed tools identified by the American Heart Association for their usefulness in assessing MetSyn risk and to help promote dietary change. Unfortunately, these tools did not address new emerging dietary factors associated with MetSyn such as low carbohydrates, reduced ultra-processed foods, and meal timing/frequency. Most of my comments below are relatively minor in nature.

Abstract:

  1. The Abstract was very well written. My only question is to include a brief rationale as to why only diet/patterns endorsed by the American Heart Association were examined.

Introduction:

  1. I think section 2 on the prevalence and economic burden fall under the Introduction. As of now, the Introduction is just a recap of the history of MetSyn definitions. Generally, the Introduction will introduce the reader to the problem at hand. I think your second section addresses that. The way it is structured now between the first and second section being different is impacting the flow of the paper.

Prevalence, Economic Burden and Causes of Metabolic Syndrome:

  1. Your first two sentences in this section need a change in perspective. I think you are setting up the change in MetSyn over time. However, those two sentences are written in present tense. I believe they need to be in past tense.
  2. There are more recent estimates from NHANES on MetSyn. You indicate that current estimates or changes over time do not exist. That is not true, exact figures do exist in NHANES. In doing just a very quick search, I was able to find the following: doi:10.1001/jama.2020.4501.
  3. Regarding the last few sentences of section 2, is it possible that hyperinsulinemia and weight gain are two co-occurring events with weight gain just being a slower responder? I know there has been a debate on which one comes first in metabolic diseases and which one to target first. Sure, they are strongly related to each other, but I think the question is what comes first. Maybe weight gain and hyperinsulinemia start around the same time, but hyperinsulinemia is quicker to the punch so to speak. Just a thought for future consideration.

Diet and Lifestyle; moving beyond weight:

  1. In the second paragraph under this section, the last sentence or two is essentially describing metabolic obesity. You have already identified the populations that are most likely to be metabolically obese. It may be worth defining this and working it in to what you already state.
  2. The sentences on lines 104-107 do not sound grammatically correct to me. I can’t make sense of what you are trying to state.
  3. In the meal timing section, it may be worth noting that there is a degree of separation between central clocks (i.e., in your brain) and peripheral clocks (i.e., in the rest of the body). In fact, the beta cells of the pancreas have their own intrinsic clock that seem to respond more to timing of food intake than light exposure, which is the primary environmental cue for setting the timing of rhythms in the central clock.

Existing tools for assessing dietary risk for Metabolic Syndrome:

  1. Why restrict your analysis of tools to just those promoted by the American Heart Association? They are not the only authority on the matter. Maybe the strongest authority on the matter, which may be the rationale, but they are not the only one. I think this needs to be clarified why you focused on their scientific statement.
  2. This may just be a formatting issue related to conversion of the paper to a PDF, but the table is practically unreadable with the spacing.

Author Response

Reviewers’ Comments:

Overall: This paper reviewed tools identified by the American Heart Association for their usefulness in assessing MetSyn risk and to help promote dietary change. Unfortunately, these tools did not address new emerging dietary factors associated with MetSyn such as low carbohydrates, reduced ultra-processed foods, and meal timing/frequency. Most of my comments below are relatively minor in nature.

Authors’ Response:

We would like to thank reviewer 1 for their time in reviewing our manuscript, and for their insightful and relevant comments. We have tried to address all comments (see below). We hope that we have interpreted all comments correctly and that subsequent changes made to the revised manuscript are to the satisfaction of Reviewer 1.

Reviewers’ Comments:

Abstract: The Abstract was very well written. My only question is to include a brief rationale as to why only diet/patterns endorsed by the American Heart Association were examined.

Authors’ Response:

Thank you to Reviewer 1 for their comment. Given we limited by word count, it is difficult to provide a detailed rationale for why we focused on the AHA recommendations. However, we agree with the Reviewer and have added a sentence in the abstract, but also provided more detail within the manuscript regarding out rationale.

Reviewers’ Comments:

Introduction: I think section 2 on the prevalence and economic burden fall under the Introduction. As of now, the Introduction is just a recap of the history of MetSyn definitions. Generally, the Introduction will introduce the reader to the problem at hand. I think your second section addresses that. The way it is structured now between the first and second section being different is impacting the flow of the paper.

Authors’ Response: Thank you to Reviewer 1 for their comment and we agree. The Prevalence and Economic Burden part of section 2 has been moved to the introduction.

Reviewers’ Comments: Your first two sentences in this section need a change in perspective. I think you are setting up the change in MetSyn over time. However, those two sentences are written in present tense. I believe they need to be in past tense.

Authors’ Response: Thank you to Reviewer 1 for their comment and we agree. We have modified the first 2 sentences to past tense 

Reviewers’ Comments: There are more recent estimates from NHANES on MetSyn. You indicate that current estimates or changes over time do not exist. That is not true, exact figures do exist in NHANES. In doing just a very quick search, I was able to find the following: doi:10.1001/jama.2020.4501.

Authors’ Response: Thank you to Reviewer 1 for their comment and we agree. We have provided more up-to-date estimates

Reviewers’ Comments: Regarding the last few sentences of section 2, is it possible that hyperinsulinemia and weight gain are two co-occurring events with weight gain just being a slower responder? I know there has been a debate on which one comes first in metabolic diseases and which one to target first. Sure, they are strongly related to each other, but I think the question is what comes first. Maybe weight gain and hyperinsulinemia start around the same time, but hyperinsulinemia is quicker to the punch so to speak. Just a thought for future consideration.

Authors’ Response: Thank you to Reviewer 1 for their comment. This is a really interesting area and as stated, an area of high debate. We agree, it is possible that both may occur at same time, with hyperinsulinemia having a more immediate effect. We believe such discussion, while interesting, is perhaps beyond the scope of this review.

Reviewers’ Comments: Diet and Lifestyle; moving beyond weight: In the second paragraph under this section, the last sentence or two is essentially describing metabolic obesity. You have already identified the populations that are most likely to be metabolically obese. It may be worth defining this and working it in to what you already state.

Authors’ Response: Thank you to Reviewer 1 for their comment and we agree. We have now defined this subgroup earlier and reworded the section where necessary

Reviewers’ Comments: The sentences on lines 104-107 do not sound grammatically correct to me. I can’t make sense of what you are trying to state.

Authors’ Response: Thank you to Reviewer 1 for their comment and we agree. We have now reworded those sentences

Reviewers’ Comments: In the meal timing section, it may be worth noting that there is a degree of separation between central clocks (i.e., in your brain) and peripheral clocks (i.e., in the rest of the body). In fact, the beta cells of the pancreas have their own intrinsic clock that seem to respond more to timing of food intake than light exposure, which is the primary environmental cue for setting the timing of rhythms in the central clock.

Authors’ Response: Thank you to Reviewer 1 for their comment and we agree. We have now provided addition detail around central and peripheral clocks, but importantly separated the two when it comes to the influence of food (or lack thereof)

Reviewers’ Comments: Existing tools for assessing dietary risk for Metabolic Syndrome: Why restrict your analysis of tools to just those promoted by the American Heart Association? They are not the only authority on the matter. Maybe the strongest authority on the matter, which may be the rationale, but they are not the only one. I think this needs to be clarified why you focused on their scientific statement.

Authors’ Response: Thank you to Reviewer 1 for their comment and we agree with your statement. While the AHA is a predominant authority figure in this area, our primary rationale was the end-users of such tools - clinicians and other members of a healthcare teams. The AHA reviewed tools against a set criteria that are relevant to routine clinical practice (i.e. can be administered quickly, are valid, and reflect ‘up-to-date’ dietary guidelines). We have added this information to the section to provide rationale for our choice

Reviewers’ Comments: This may just be a formatting issue related to conversion of the paper to a PDF, but the table is practically unreadable with the spacing.

Authors’ Response: Thank you to Reviewer 1 for their comment and we apologise for this formatting error. We have re-formatted to improve readability.

Reviewer 2 Report

The paper examines various dietary assessment tools (MEDAS, REAP-S, and “Starting the Conversation”) for metabolic syndrome and investigates whether they align with recent developments and evidence related to metabolic syndrome, in particular (the restriction of) carbohydrates, (the minimization of) ultra-processed food, and (the modification of) eating times and frequency.

Authors support that even normal-weighted individuals with metabolic syndrome should be targeted for intervention (in particular, increased physical activity).

Overall, the paper is well written and presented. My main comment is about the results, as presented in the table, and their analysis. I am not sure if the conclusions of the authors are represented well in the manuscript. Also, some additional comments could be made. For example, looking at the table, it seems that “Starting the Conversation” significantly covers both ultra-processed foods as well as eating times and frequency. However, for REAP-S the results are different, with most cases either lacking or missing details. Additionally, there are a lot of cases where more details are required, which could in turn change the degree of relevance of each tool to each of the three categories.

The results are presented in a table; however, the table’s appearance could be improved. In its current state, it spans more than 4 pages, and is not easy to read. Various improvements could be made to re-organize and shorten the table (e.g., replace “Mode detailed required” with another symbol such as “?” and explain it in the caption). It is important to note, however, that this is only about presentation; the content and information of the table is fine.

Author Response

Reviewer 2

Reviewers’ Comments: The paper examines various dietary assessment tools (MEDAS, REAP-S, and “Starting the Conversation”) for metabolic syndrome and investigates whether they align with recent developments and evidence related to metabolic syndrome, in particular (the restriction of) carbohydrates, (the minimization of) ultra-processed food, and (the modification of) eating times and frequency. Authors support that even normal-weighted individuals with metabolic syndrome should be targeted for intervention (in particular, increased physical activity).

Authors’ Response: We would like to thank reviewer 2 for their time in reviewing our manuscript, and for their insightful and relevant comments. We have tried to address all comments (see below). We hope that we have interpreted all comments correctly and that subsequent changes made to the revised manuscript are to the satisfaction of Reviewer 2.

Reviewers’ Comments: Overall, the paper is well written and presented. My main comment is about the results, as presented in the table, and their analysis. I am not sure if the conclusions of the authors are represented well in the manuscript. Also, some additional comments could be made. For example, looking at the table, it seems that “Starting the Conversation” significantly covers both ultra-processed foods as well as eating times and frequency. However, for REAP-S the results are different, with most cases either lacking or missing details. Additionally, there are a lot of cases where more details are required, which could in turn change the degree of relevance of each tool to each of the three categories.

Authors’ Response: Thank you to Reviewer 2 for their comment and we apologize that a formatting error made the table difficult to read and interpret.  In addition to re-formatting the table for readability, we have added further discussion (before Table 1) which briefly summarises your key points mentioned above. As a result, we moved other parts of discussion to improve the flow of the manuscript.

Reviewers’ Comments: The results are presented in a table; however, the table’s appearance could be improved. In its current state, it spans more than 4 pages, and is not easy to read. Various improvements could be made to re-organize and shorten the table (e.g., replace “Mode detailed required” with another symbol such as “?” and explain it in the caption). It is important to note, however, that this is only about presentation; the content and information of the table is fine.

Authors’ Response: Thank you to Reviewer 2 for their comment and we apologize for this formatting error. We have re-formatted to improve readability.

Round 2

Reviewer 2 Report

The authors have addressed my comments. However, some additional emphasis on the main findings/thesis of the paper could be given. The main outcome of the work is presented in Table 1 and section 4.4. It would be interesting to expand this section a bit more.

Author Response

Reviewers’ Comments: The authors have addressed my comments. However, some additional emphasis on the main findings/thesis of the paper could be given. The main outcome of the work is presented in Table 1 and section 4.4. It would be interesting to expand this section a bit more.

Authors’ Response: We would like to thank reviewer 2 for their time in reviewing our manuscript, and for their additional comments. We have tried to address comments by expanding section 4.4 of the paper thereby giving greater emphasis to the main findings. We hope that we have interpreted all comments correctly and that subsequent changes made to the revised manuscript are to the satisfaction of Reviewer 2.